# TWEAK mediates inflammation in experimental atopic dermatitis and psoriasis

Daniel Sidler[1], Ping Wu[2], Rana Herro[1], Meike Claus[1], Dennis Wolf[3], Yuko Kawakami[4], Toshiaki Kawakami[4], Linda Burkly[2] & Michael Croft[1]

Atopic dermatitis (AD) and psoriasis are driven by alternate type 2 and type 17 immune responses, but some proteins might be critical to both diseases. Here we show that a deficiency of the TNF superfamily molecule TWEAK (TNFSF12) in mice results in defective maintenance of AD-specific T helper type 2 (Th2) and psoriasis-specific Th17 cells in the skin, and impaired expression of disease-characteristic chemokines and cytokines, such as CCL17 and TSLP in AD, and CCL20 and IL-19 in psoriasis. The TWEAK receptor, Fn14, is upregulated in keratinocytes and dermal fibroblasts, and TWEAK induces these cytokines and chemokines alone and in synergy with the signature T helper cytokines of either disease, IL-13 and IL-17. Furthermore, subcutaneous injection of recombinant TWEAK into naive mice induces cutaneous inflammation with histological and molecular signs of both diseases. TWEAK is therefore a critical contributor to skin inflammation and a possible therapeutic target in AD and psoriasis.

[1] Division of Immune Regulation, La Jolla Institute for Allergy and Immunology, La Jolla, California 92037, USA. [2] Department of Immunology, Biogen, 115 Broadway, Cambridge, Massachusetts 02142, USA. [3] Inflammation Biology, La Jolla Institute for Allergy and Immunology, La Jolla, California 92037, USA. [4] Cell Biology, La Jolla Institute for Allergy and Immunology, La Jolla, California 92037, USA. Correspondence and requests for materials should be addressed to L.B. (email: linda.burkly@biogen.com) or to M.C. (email: mick@lji.org).

Atopic dermatitis (AD) and psoriasis are inflammatory disorders of the skin that cause substantial morbidity[1,2]. Although most cases are mild, some patients experience severe or widespread disease that can be physically, socially and emotionally debilitating. Allergens are the primary cause of AD, whereas the triggers for psoriasis are not clear. Each disease is probably multifactorial, with interplay of genetic predisposition and environmental factors[3–6]. Both diseases feature infiltration of immune cells, dermal changes, and epidermal acanthosis and hyperkeratosis and parakeratosis. Keratinocytes and dermal fibroblasts are of central importance to the pathogenesis of AD and psoriasis by responding to external and internal danger signals and expressing proinflammatory cytokines and chemokines, which attract cells of the innate and adaptive immune systems[5,7,8]. Despite these commonalities, AD and psoriasis are believed to develop through distinct immunological programs and may engage mutually biased immune signatures, with type 2 immune responses dominating AD[9] and type 17 immune responses dominating psoriasis[10–12]. A clinical study demonstrated that blocking IL-4 and IL-13 is beneficial for patients with treatment-resistant AD[13], whereas inhibition of IL-23 p19 (ref. 14) or IL-17A (refs 15–17) is highly effective in treating psoriasis patients. Regardless of the success of these current treatments, whether they will be completely effective at reducing all symptoms or will result in long-lasting control is unclear. Continuing efforts to better understand the molecular basis of these diseases may lead to new anti-inflammatory drugs that could be either stand-alone treatments, or be combined with existing therapies.

Many members of the TNF superfamily are emerging as important modulators of immune-mediated disorders[18–21]. Blockers of TNF (TNFSF2) are already approved as anti-inflammatory agents for plaque psoriasis. Another family member, TWEAK (TNFSF12), can be expressed by myeloid, stromal and epithelial cells of various tissues and binds to its specific receptor Fn14 (TNFRSF12A) that is expressed on a range of target cells[22–25]. Fn14 has a low level of ubiquitous expression on many cells, but is induced under conditions of stress or inflammation, one of the main regulatory mechanisms for TWEAK/Fn14 action. Fn14 has been shown to induce canonical and non-canonical NFκB signalling and leads to the expression of a number of inflammatory mediators in various cell types that can contribute to tissue inflammation and regenerative activities[22–25]. TWEAK has been shown to participate in several inflammatory conditions in mice, including ischemia and reperfusion injury and neurodegeneration, inflammatory bowel disease, hepatitis, arthritis, and lupus-like kidney disease.

In the current study, we show a major role of the TWEAK/Fn14 pathway in immune-mediated skin inflammation using gene-deficient mice in clinically relevant models of AD and psoriasis. Furthermore, subcutaneus injection of TWEAK in naive animals is sufficient to induce localized skin inflammation with histological and molecular features of both AD and psoriasis. We identify several inflammatory mediators that are controlled by TWEAK in vivo, and show that epidermal keratinocytes and dermal fibroblasts are cellular targets of TWEAK that contribute to production of these mediators. These include proinflammatory chemokines common to AD and psoriasis, namely CCL2, CCL5 and CCL7, that mediate recruitment of monocytes, T cells, and eosinophils. In addition, TWEAK can enhance production of more disease-specific chemokines and cytokines, such as CCL17, CCL20, TSLP and IL-19 that further orchestrate skin inflammation. We conclude that TWEAK is a critical mediator of the pathogenesis of both disorders.

## Results

**TWEAK deficiency limits severity of atopic dermatitis**. We first assessed the expression of TWEAK and Fn14 in the skin in experimental AD. This was induced by repetitive topical exposure to house dust mite (HDM) allergen and staphylococcal enterotoxin B (SEB)[26,27], both implicated in the pathogenesis of AD in humans[28–30]. Dermatitis develops in a T-cell-dependent manner[31] with involvement of type 2 cytokines, including TSLP[32], that are characteristic of human AD. Transcripts for both TWEAK and Fn14 were detectable in the skin of naive animals but their expression was substantially increased by 5–10-fold in AD lesions (Fig. 1a,b). In agreement, Fn14 protein was faintly detectable in naive skin specimens and upregulated in AD lesions (Supplementary Fig. 1a,b).

We then tested the requirement for TWEAK using gene-deficient mice. WT mice developed severe skin disease with characteristic clinical features of AD, including erythema, bleeding, excoriation and scaling. Histologically, acanthosis and dermal fibrosis were evident, reflected by thickening of the epidermal and dermal layers and excessive collagen deposition in the dermis (Fig. 1c–e). In contrast, TWEAK-deficient animals developed substantially reduced disease when analysed for the same end points. Of note, and in agreement with these findings, IL-13 transcripts in the skin were significantly reduced in TWEAK-deficient animals at the end of the experiment when compared to WT littermates (Fig. 1f). However, at earlier time points (day 7) IL-13 mRNA expression was equivalent in the skin and IL-13$^+$ CD4$^+$ T cells were detected at a normal frequency in the skin draining lymph nodes (Fig. 1g). These data indicates that TWEAK-deficient animals mounted an initial Th2 response, perhaps not surprising as T cells do not express Fn14, and suggests that TWEAK indirectly controlled maintenance of the Th2 response in the skin.

We further tested the requirement for TWEAK in AD pathogenesis in NC/Nga mice (Supplementary Fig. 2), a strain that develops a more severe AD phenotype after exposure to HDM[26,33]. Indeed, these mice exhibited greater histological manifestations compared to C57BL/6 animals, including dramatic hyperkeratosis, mononuclear and lymphocytic infiltration and scaling, as well as pronounced dermal fibrosis involving the subcutaneous adipose tissue and reaching subdermal muscular layers. Confirming the data from TWEAK-deficient C57BL/6 animals, prophylactic treatment with a neutralizing TWEAK antibody significantly ameliorated clinical and histological skin disease in NC/Nga animals.

**TWEAK and Fn14 are active in experimental psoriasis**. We then assessed the expression of TWEAK and Fn14 in the skin in experimental psoriasis induced by the TLR7/8 agonist Imiquimod (IMQ) that promotes histological features seen in human psoriasis, including acanthosis, papillomatosis and parakeratosis[34]. Skin inflammation in this model is dependent on the type 17-associated cytokines IL-17A and IL-23 (ref. 35) that are also produced in human psoriasis. Again, expression of both TWEAK and Fn14 were increased up to 20-fold in psoriatic skin tissue (Fig. 2a,b), and the protein was readily detectable in keratinocytes of the stratum granulosum in psoriatic skin lesions as well as in the dermis (Supplementary Fig. 1a,c).

IMQ-induced severe dermatitis in WT animals with pronounced acanthosis, parakeratosis and papillomatosis; however, these features were strongly reduced in TWEAK-deficient animals. In line, there was markedly less epidermal thickness in TWEAK-deficient animals, including a reduction in hyperkeratosis (Fig. 2c–e). The dermal contribution in this disease model is minor (Fig. 2c,e), in contrast to the AD model where both epidermal and

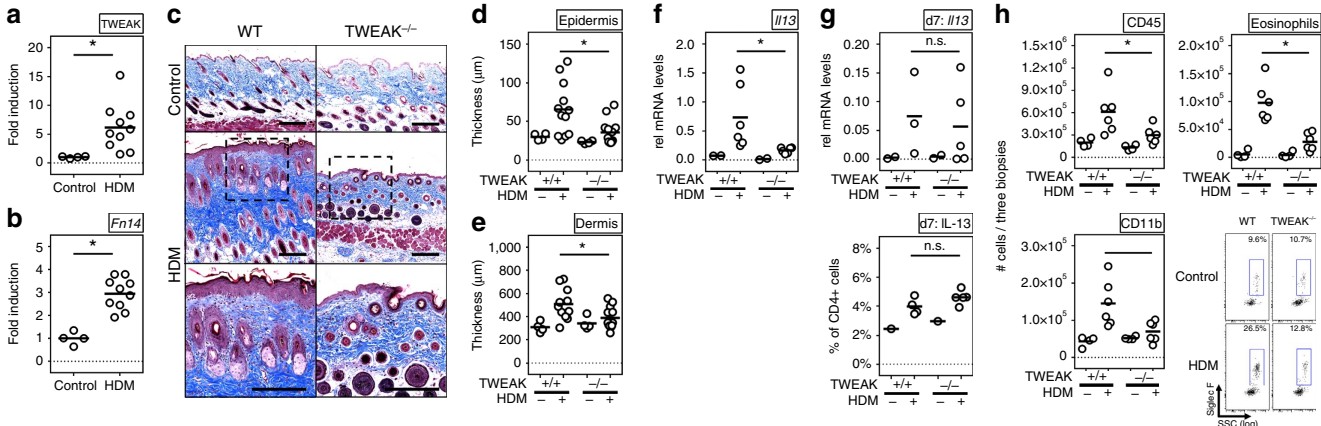

**Figure 1 | TWEAK controls disease activity in HDM-induced AD.** WT and TWEAK-deficient mice were treated epicutaneously with HDM on the shaved back for 23 days to induce AD-like disease. (**a,b**) TWEAK and Fn14 mRNA fold induction in WT mice relative to expression in skin from naive animals (Control). Combined results from three experiments. Each data point represents one individual mouse. (**c**) Representative Masson's trichrome staining of skin sections from WT and TWEAK$^{-/-}$ animals treated with HDM or PBS control. Bar represents 200 μm. (**d,e**) Epidermal and dermal thickness measured on trichrome-stained sections. Results combined from three independent experiments. Each data point represents one animal. (**f**) IL-13 mRNA in skin lesions on day 23. Combined from two experiments. (**g**) IL-13 mRNA in skin biopsies (top), and frequency of IL-13 positive CD4$^+$ T cells in skin-draining lymph nodes on day 7 (bottom). Four mice per group. (**h**) Total numbers of CD45$^+$ cells (left top), CD45$^+$CD11b$^+$ monocytes (left bottom), and SigF$^+$CD45$^+$CD11b$^+$Ly6G$^-$ eosinophils (right top and bottom) in skin biopsies on day 23. Combined results from two experiments, with each data point one animal. Bars represent mean values. The Mann–Whitney U-test was performed to compare the groups. Asterisks indicate $P < 0.05$. n.s. indicates nonsignificant differences.

dermal changes are evident. Intriguingly, the expression of IL-17A mRNA in the skin, and the frequency of Th17 cells in the draining lymph nodes, was strikingly lower in the absence of TWEAK at the end of the experiment (Fig. 2f), whereas mRNA for IL-17A and CD4$^+$ T cells expressing IL-17 were not affected at early times (Fig. 2g). This suggests that a Th17 response was initiated in the absence of TWEAK, but this could not be maintained in the skin over time. In summary, these results indicate that TWEAK is instrumental in the development of the characteristic histological features of both AD and psoriasis, and the absence of TWEAK/ Fn14 interactions can limit the adaptive immune response in the skin regardless of the type 2 phenotype characteristic of AD or the type 17 phenotype characteristic of psoriasis.

**TWEAK regulates immune cell infiltration in skin lesions**. We next analysed the cellular infiltrates of lesional skin. In the AD model, a mononuclear (CD45$^+$CD11b$^+$) and eosinophil dominant infiltrate was evident in dermatitis lesions (Fig. 1h, Supplementary Fig. 5 for gating strategy). Skin infiltrates appeared after 4 days and accumulated thereafter until full establishment of disease. In the psoriasis model, a prominent mononuclear and transient granulocytic infiltrate was evident in lesions within the first days of disease (Fig. 2h). Eosinophils were virtually absent in IMQ-induced disease because of the lack of a Th2 response. Interestingly, in both models, TWEAK-deficient animals showed significantly reduced numbers of infiltrating CD45$^+$ cells compared to WT littermates, and reduced numbers of CD11b$^+$ myeloid cells were evident in skin lesions of HDM-treated (Fig. 1h) and IMQ-treated (Fig. 2h) mice. Eosinophils were also almost absent in HDM-primed TWEAK-deficient animals (Fig. 1h). These data suggest that TWEAK directly or indirectly controls the recruitment and/or maintenance of innate inflammatory cells in the skin in addition to regulating the persistent accumulation of Th2 and Th17 cells.

**TWEAK is sufficient to induce skin inflammation**. To further understand the mechanism of action of TWEAK, we asked whether the recombinant protein alone was able to drive skin inflammation in the absence of any other inflammatory stimulus. For this purpose, we injected naive animals with rTWEAK as an Ig fusion protein, or an irrelevant control Ig, via the subcutaneous route. Strikingly, several injections of rTWEAK over 2 weeks in the rostral regions of the back induced a clinically defined dermatitis localized to the region of injection with features of both AD and psoriasis, although rTWEAK alone did not entirely mimic the phenotypes seen in the HDM and IMQ models (Fig. 3a). A single dose of rTWEAK had a rapid effect on epidermal hyperplasia (Fig. 3b, day 4), although this effect increased with further doses over time (Fig. 3b, days 10 and 14). In contrast, dermal reactions only became evident after repetitive administrations (Fig. 3c, days 10 and 14, and Fig. 3a), thus in total resembling a mix of histology observed in the AD and psoriasis models. Importantly, Fn14-deficient animals were completely protected from skin inflammation and epidermal and dermal hyperplasia, indicating a specific activity of rTWEAK via its receptor Fn14 (Supplementary Fig. 3). Analysis of infiltrates revealed that rTWEAK induced monocyte accumulation and moderate eosinophilia in the skin (Fig. 3d). Thus, rTWEAK is sufficient to independently promote skin inflammation and at least some activities associated with AD and psoriasis.

**TWEAK controls production of chemokines in the skin**. Based on the data obtained in the absence of TWEAK in the AD and psoriasis models, and with injection of rTWEAK, this led to the hypothesis that one major activity of TWEAK/Fn14 could be to control the migration of immune cells into the skin or their retention in the skin after inflammation is initiated. We therefore assayed for chemokine gene signatures in animals injected with rTWEAK when compared to control Ig injected WT animals, or naive animals, or rTWEAK injected Fn14-deficient animals. By RNA sequencing analysis, we identified expression differences in multiple members of the CC- and CXC-chemokine families that were significantly upregulated by TWEAK/Fn14 interactions (Fig. 4a).

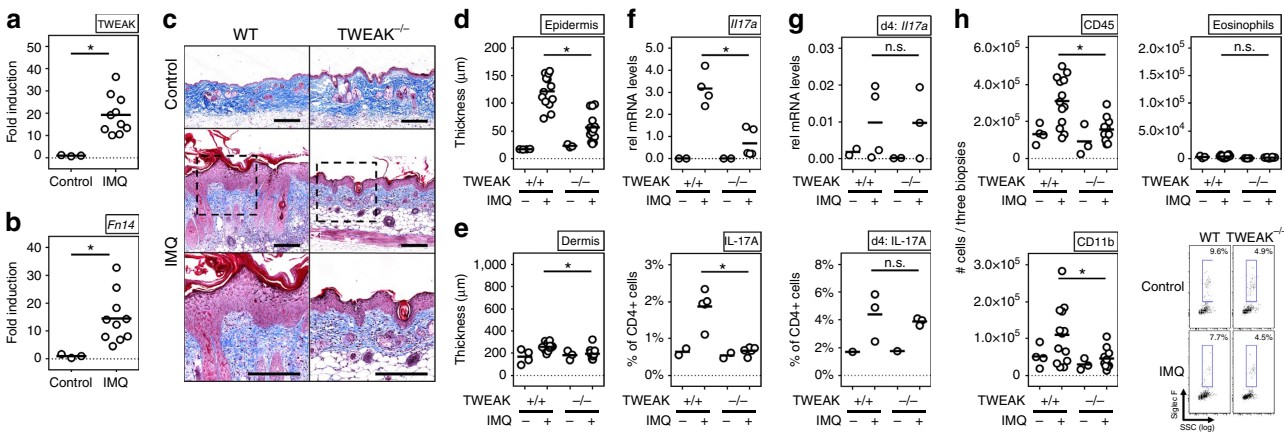

**Figure 2 | Reduced disease severity in IMQ-induced psoriasis in the absence of TWEAK.** WT and TWEAK-deficient mice were treated epicutaneously on the shaved back with IMQ for 8 days to induce psoriasis-like disease. (**a,b**) TWEAK and Fn14 mRNA fold induction in WT mice relative to expression in skin from naive animals (control). Combined results from three experiments. Each data point an individual mouse. (**c**) Representative Masson's trichrome staining of skin sections. Scale bar, 200 μm. (**d,e**) Epidermal and dermal thickness on day 8 measured on trichrome-stained sections. (**f,g**) IL-17 mRNA in skin lesions and frequency of IL-17 positive CD4+ T cells in skin-draining lymph nodes on day 8 (**f**) or day 4 (**g**). Each datapoint one mouse. (**h**) Total numbers of CD45+ cells (left top), CD45+CD11b+ monocytes (left bottom) and SigF+CD45+CD11b+Ly6G− eosinophils (right) from skin biopsies obtained on day 8. Combined results from three experiments. Each data point one animal. Bars represent mean values. The Mann–Whitney U-test was performed to compare the groups. Asterisks indicate P < 0.05. n.s. indicates nonsignificant differences.

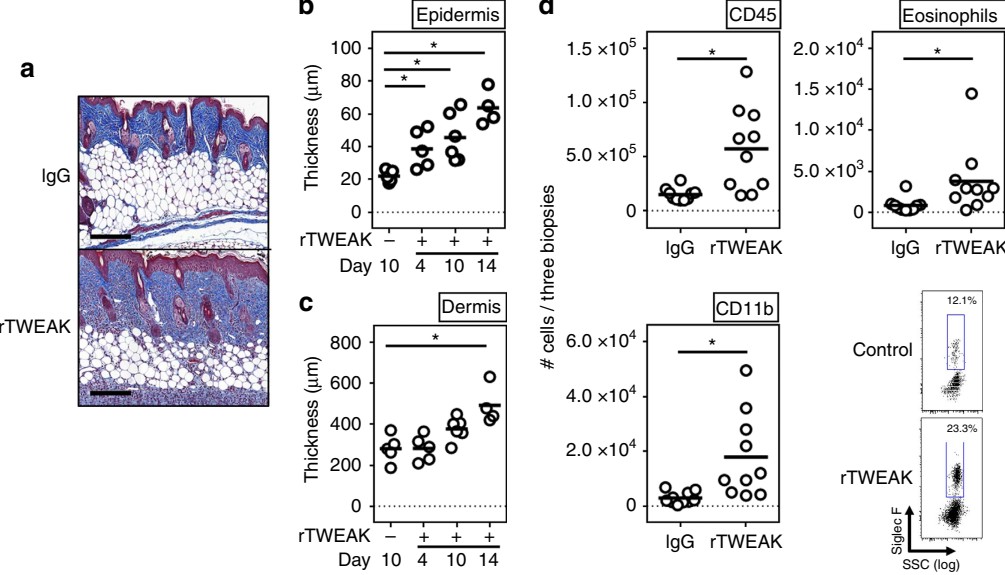

**Figure 3 | rTWEAK induces skin inflammation.** (**a,b**) Naive WT mice were injected with 75 μg mouse rTWEAK or IgG, administered s.c. either four times during a 14 day period (**a**), or once with analysis on day 4, or twice weekly with analysis on day 10 (**b,c**). (**a**) Masson's trichrome staining of representative skin sections on day 14. Scale bar, 200 μm. (**b,c**) Epidermal and dermal thickness on days 4, 10 and 14. Results are combined from two independent experiments. (**d**) Absolute number of skin-infiltrating cells measured by flow cytometry on day 4, and representative flow plot of skin-infiltrating eosinophils (SigF+CD45+CD11b+Ly6G− cells). Combined from two experiments. Each data point one animal. Bars represent mean values. The ANOVA test with Bonferroni Correction was performed to compare the groups. Asterisks indicate P < 0.05. n.s. indicates nonsignificant differences. ANOVA, analysis of variance.

We then focused on five chemokines thought to be of primary importance to skin inflammation for further analysis. CCL2, CCL5 and CCL7 have been associated with skin inflammatory responses in general, regulating attraction of monocytes/macrophages, T cells or eosinophils, while CCL17 and CCL20 have been shown to be primarily involved in Th2 responses and AD, or Th17 responses and psoriasis, respectively[36,37]. By quantitative PCR analysis, we confirmed strong upregulation of these chemokines in skin specimens after a single injection of rTWEAK

(Fig. 4b). To test the importance of several of these molecules in TWEAK-mediated skin inflammation, we blocked CCL2, CCL5 and CCL7 together, by injecting neutralizing antibodies at the time of rTWEAK injection. Indeed, this significantly ameliorated skin infiltrates, epidermal thickening and the dermal reaction induced by rTWEAK (Fig. 4c–f).

Most importantly, TWEAK-deficient animals displayed a marked reduction in expression of chemokines in the skin in both the AD and psoriasis models correlating with their

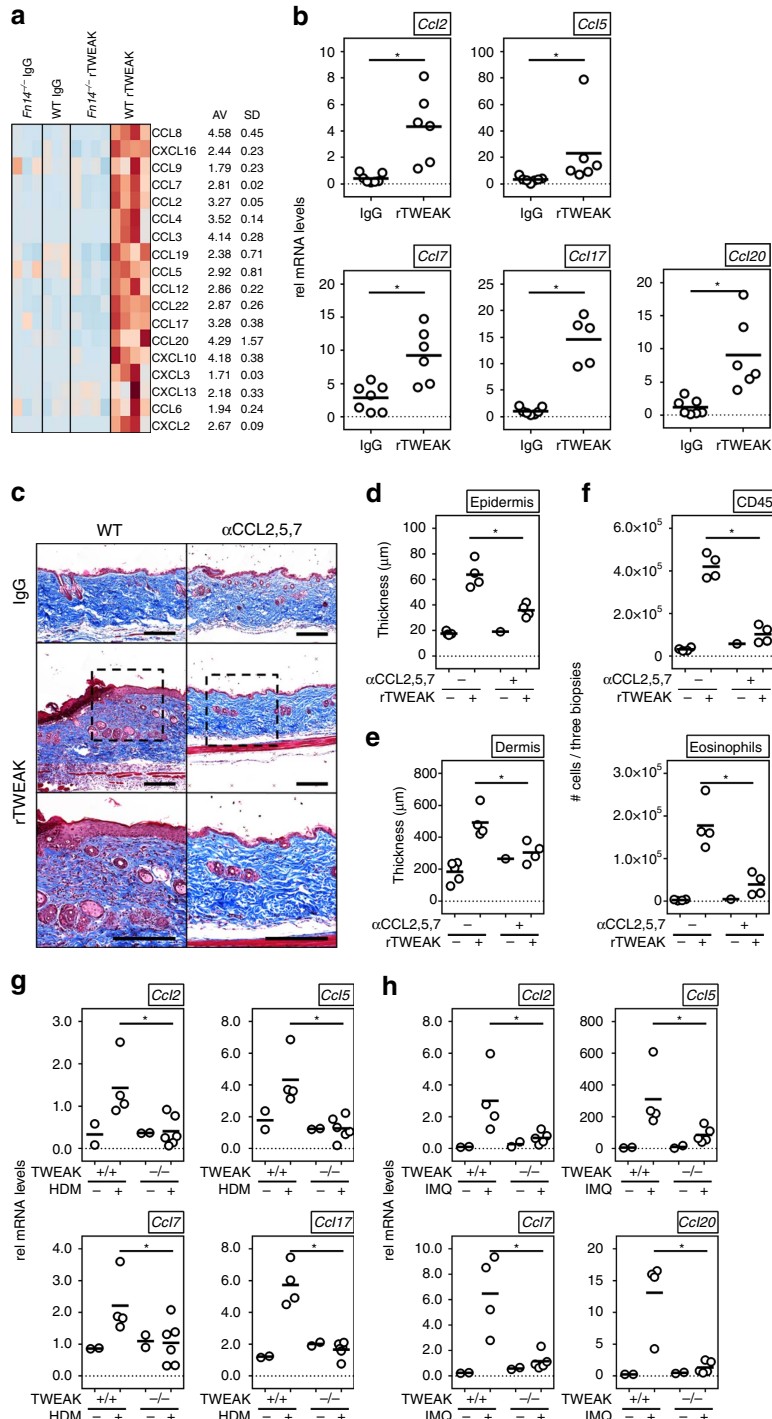

**Figure 4 | TWEAK regulates chemokine expression in the skin.** (**a**,**b**) Naive WT or Fn14-deficient mice were injected with rTWEAK or IgG, administered once s.c. After 4 days expression of mRNA for indicated chemokines was measured in skin biopsies. (**a**, left) RNAseq heatmap of chemokine expression from 3 to 4 animals/group. (right) Mean (AV) fold increase and SD calculated from RNAseq data of the comparisons of rTWEAK into WT versus Fn-deficient mice, rTWEAK versus IgG into WT mice, and rTWEAK versus no injection into WT mice. (**b**) qPCR mRNA expression of indicated chemokines relative to GAPDH. Combined results from two experiments. Each data point one mouse. (**c**–**f**) naive WT mice were injected with rTWEAK or IgG (s.c.) twice weekly over 14 days as in Fig. 3a, with either a combination of blocking antibodies to CCL2, CCL5 and CCL7, or a combination of control antibodies (i.p.). Combined results from two experiments. (**c**) Masson's trichrome staining of representative skin sections. Scale bar, 200 μm. (**d**,**e**) Quantification of epidermal and dermal thickness. (**f**) Total numbers of CD45+ cells and SigF+CD45+CD11b+Ly6G− eosinophils from skin biopsies. (**g**,**h**) WT and TWEAK-deficient animals were treated with HDM for 23 days (**g**) and IMQ for 8 days (**h**) and mRNA for indicated chemokines assessed in skin, relative to GAPDH expression. Combined from two experiments. Each data point one mouse. Bars represent mean values. The Mann–Whitney $U$-test was performed to compare the groups. Asterisks indicate $P < 0.05$. qPCR, quantitative PCR.

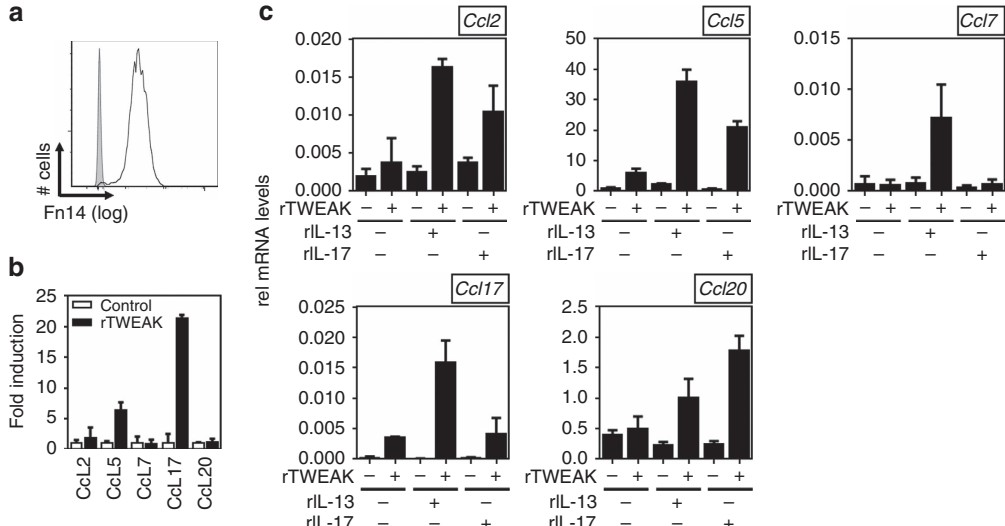

**Figure 5 | TWEAK synergizes with IL-13 and IL-17A to induce chemokines in human keratinocytes.** (**a**) Basal surface expression of Fn14 (line) on NHEK cells relative to Isotype staining (grey). (**b,c**) NHEK cells were stimulated for 48 h with rTWEAK, rIL-13, rIL-17 or their combination. mRNA expression of indicated chemokines was assessed relative to GAPDH. Means of triplicate cultures with standard deviations. Values were compared using Student's *t*-test. One out of three independent experiments.

induction in animals injected with rTWEAK (Fig. 4g,h). We confirmed that endogenous TWEAK was required for maximal expression of CCL2, CCL5 and CCL7, and furthermore, we found defective production of CCL17 and CCL20 in mice lacking TWEAK in the AD model and the psoriasis model, respectively (Fig. 4g,h). This indicates that TWEAK-deficient animals are incapable of inducing a sustained inflammatory chemokine milieu within the skin which may explain why they cannot maintain disease-mediating immune cells in the context of AD and psoriasis reactions.

**TWEAK induces chemokine expression in keratinocytes.** One likely source of chemokines in the skin is the keratinocyte. We visualized Fn14 in normal human epidermal keratinocytes (Fig. 5a) and a murine keratinocyte cell line (Supplementary Fig. 4a) by flow cytometry. We first tested whether TWEAK could induce production in human keratinocytes of the five chemokines that we choose as the focus of our further studies. We also tested whether TWEAK might function in synergy with the signature cytokines of AD and psoriasis, IL-13 and IL-17A, whose receptors are also expressed on keratinocytes. rTWEAK alone had mild/moderate activity in inducing CCL2, 5, and 17 (Fig. 5b). Interestingly, enhanced expression of CCL2 and CCL5 was observed when rTWEAK was combined with either recombinant IL-13 or IL-17A, whereas TWEAK-induced CCL17 was augmented by rIL-13 but not rIL-17 in line with CCL17 being associated with type 2 responses (Fig. 4c). CCL7 and CCL20 were not induced by any of these molecules in isolation. However, CCL7 was produced when rTWEAK was combined with rIL-13. CCL20 was upregulated when rTWEAK acted together with either cytokine, although in line with CCL20 being more associated with type 17 responses, a greater effect was seen with TWEAK and IL-17 (Fig. 4c). These findings were also replicated to a large extent with the murine keratinocyte cell line PAM212. Although these cells were less responsive to TWEAK stimulation alone, synergistic activities were again seen with IL-13 and/or IL-17 in driving production of all of these chemokines, including specifically augmenting CCL17 with rIL-13 and CCL20 with rIL-17 (Supplementary Fig. 4). These data, combined with our *in vivo* mouse results, show that TWEAK can

directly promote chemokines that are common to both AD and psoriasis, as well as can enhance expression of chemokines that are more specifically associated with the type 2 and type 17 responses underlying these diseases.

**TWEAK induces cytokines associated with AD and psoriasis.** In addition, we determined whether TWEAK controlled expression of other molecules that are associated with either AD or psoriasis. Most interestingly, we observed that a single injection of rTWEAK into naive animals over 4 days induced expression of TSLP and IL-19 (Fig. 6a,c). TSLP has been found upregulated in skin lesions of AD patients[38], and IL-19 has been shown to be upregulated in the skin of psoriasis patients[39,40] (Fig. 6a,c). Confirming that endogenous TWEAK activity had a function in TSLP and IL-19 production *in vivo*, gene-deficient mice exhibited reduced expression of these molecules in the skin in the AD and psoriasis models, respectively (Fig. 6b,d). Both TSLP and IL-19 have been reported to be products of keratinocytes and fibroblasts. Reinforcing the notion that keratinocytes are a likely primary target of TWEAK, we found that rTWEAK promoted the expression of TSLP in normal human epidermal keratinocytes in a dose-dependent manner. rTWEAK also induced some TSLP in normal human primary dermal fibroblasts, albeit at much lower levels than seen in keratinocytes (Fig. 6e). In contrast, keratinocytes showed very limited capacity to express IL-19 constitutively or in response to rTWEAK, but dermal fibroblasts responded to rTWEAK by making IL-19 (Fig. 6f). Together with the prior chemokine data, these results show that TWEAK can specifically augment activities from several skin structural cell types that are thought to have key roles in the pathogenesis of both AD and psoriasis.

**Upregulation of Fn14 occurs in human AD and psoriasis.** Substantiating the notion that our results might have application to human disease, a growing body of recent literature shows that TWEAK and its receptor Fn14 are present in the skin and one or both molecules are significantly upregulated in lesions from AD and psoriasis patients or in serum[41–44]. Indeed, we performed a gene set enrichment analysis from a recent comparitive study of

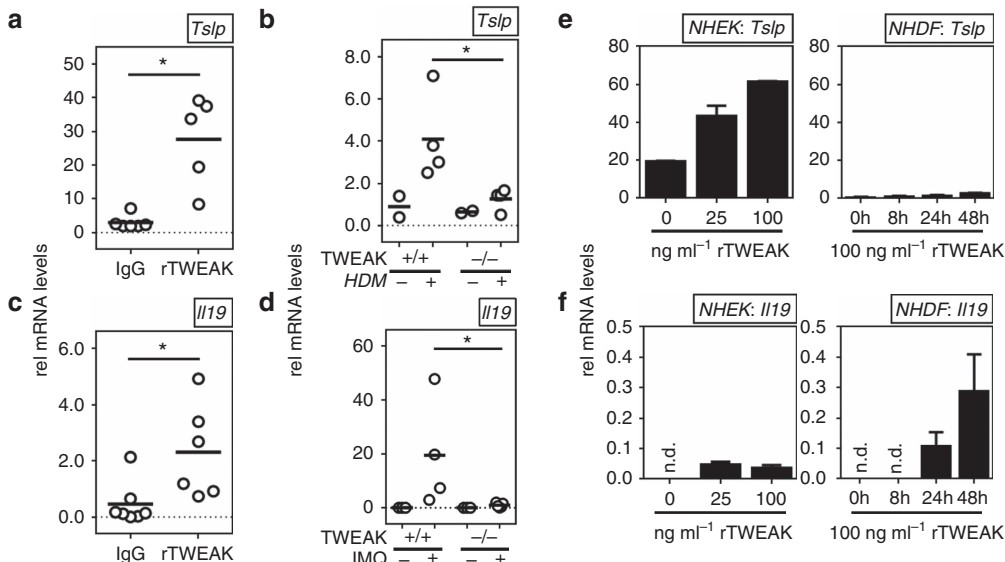

**Figure 6 | TWEAK promotes cytokines associated with AD and psoriasis.** (**a**,**c**) Naive WT mice were injected with rTWEAK or IgG administered once s.c. After 4 days expression of mRNA for TSLP and IL-19 was measured in skin biopsies. Data relative to GAPDH. Combined results from two experiments. (**b**,**d**) WT and TWEAK-deficient animals were treated with HDM for 4 days (**b**) or IMQ for 8 days (**d**) and mRNA for TSLP and IL-19, respectively, assessed in skin, relative to GAPDH expression. One out of three experiments shown. (**e**,**f**) NHEK cells (left) or NHDF cells (right) were stimulated for 48 h or the indicated times with rTWEAK. mRNA expression of TSLP (**e**) and IL-19 (**f**) was assessed relative to GAPDH. Each data point in **a**–**d** represents one mouse. Bars represent mean values. The Mann–Whitney $U$-test was performed to compare the groups. Bars in (**e**,**f**) indicate mean values of triplicates and s.d. Values were compared using Student's $t$-test. One out of three independent experiments. Asterisks indicate $P < 0.05$.

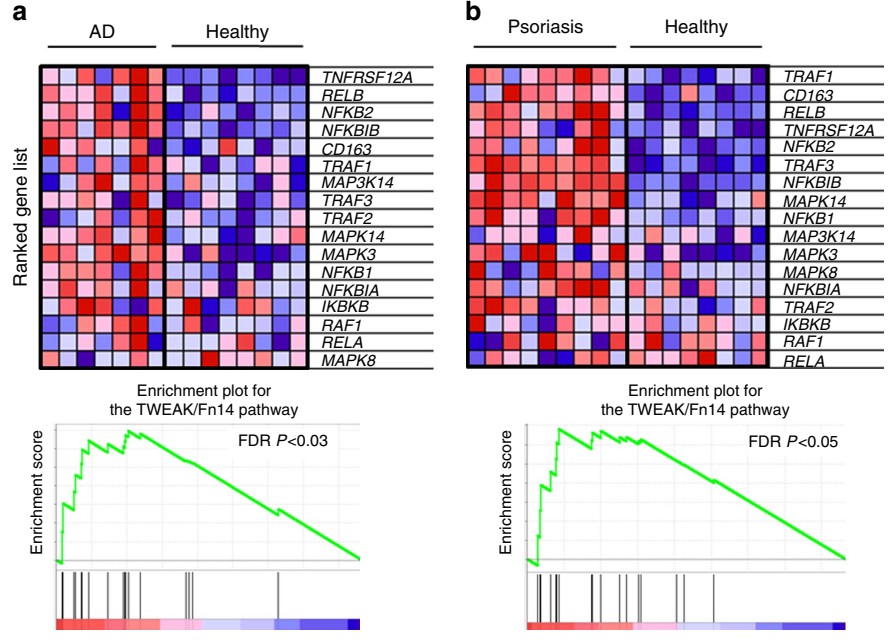

**Figure 7 | Gene set enrichment for the TWEAK/Fn14 pathway in AD and psoriasis.** Gene set enrichment analysis of skin biopsies from healthy ($n = 8$), AD ($n = 7$), and psoriasis patients ($n = 9$). GenePattern 2.0 (ref. 74) was used to process RNA array data for gene set enrichment analysis[75] which was run with the default settings (100 iterations, weighted). A FDR $P$ value $< 0.05$ was considered significant. Fn14 (TNFRSF12A) and intracellular signalling and effector molecules are significantly enriched among the induced transcripts from AD (**a**) and psoriasis lesions (**b**) when compared to healthy control specimens (FDR $P < 0.05$ in both comparisons). FDR, false discovery rate.

skin specimens from AD and psoriasis patients[45] and confirmed that Fn14 and a number of signalling molecules associated with its activity are strongly increased in lesions of both diseases when compared to specimens from healthy controls (Fig. 7). Collectively, the data support a role for the TWEAK/Fn14 axis in the pathogenesis of these human disorders.

## Discussion

The epithelial cell barrier contributes to local immune homoeostasis and efforts to modulate the response of these cells to inflammatory signals are ongoing. In particular, the cytokines IL-17A and IL-13 that are current targets of clinical therapy for the skin inflammatory diseases psoriasis and AD have receptors

expressed on keratinocytes, and it has been postulated that the efficacy of neutralizing these cytokines in suppressing disease symptoms is in part related to dampening keratinocyte activity. We now identify keratinocytes, as well as dermal fibroblasts, as cells that can upregulate a number of chemoattractive and proinflammatory factors that are associated with AD and psoriasis in response to TWEAK, and that TWEAK can also synergize with signals from IL-17A or IL-13 to further upregulate some of these molecules. Compellingly, a genetic deficiency in TWEAK in mice, or neutralizing this molecule, ameliorated skin inflammatory activity characteristic of both AD and psoriasis, implying that TWEAK may have a key role in the pathogenesis of both human disorders.

The notion that AD and psoriasis are mutually exclusive diseases in humans stems from an immunological standpoint from experimental and clinical results suggesting that they are driven by separate lineages of T cells, as well as other immune cell types, that contribute Th2 versus Th17 cytokines, respectively. However, given that these individual T-cell populations and characteristic cytokines are key drivers of the phenotype and pathology of each disease, it is still likely that there are inflammatory factors that will be common to both diseases that will aid, regulate or synergize with, these individual T-cell populations or cytokines. Our data clearly demonstrate that TWEAK is important for maximal disease in two well-established models of AD and psoriasis that display the Th2 (IL-13) or Th17 (IL-17) phenotypes. We validate in our studies that these cytokines and resultant immune cell phenotype (presence or absence of eosinophils) are exclusive to each of our models, but importantly show that in either case TWEAK and its receptor are upregulated at the same time. The rationale for why TWEAK and Fn14 can participate in so-called mutually exclusive skin diseases is then that these molecules are not restricted to a particular lineage of cells, they do not control the generation of these lineages of cells, and they do not rely on these specialized lineages for their production or upregulation. TWEAK alone cannot promote full AD or psoriasis phenotypes but it can promote chemokines common to both AD and psoriasis from keratinocytes and fibroblasts, as well as characteristic cytokines of both diseases such as TSLP and IL-19. As importantly, TWEAK can synergize with the disease-specific cytokines IL-13 and IL-17 in these activities, further explaining why this one pathway can be important for pathology in two diseases thought to be fundamentally different from one another.

A primary effect of TWEAK in the skin, revealed by subcutaneous injection of the recombinant protein, was to promote expression of multiple chemokines. Studies over the past 10 years have solidified the notion that chemokines are pivotal in determining the localization of immune cells, maintenance of normal immune system function, and defense against pathogens[46]. Moreover it is evident that dysregulated expression of chemokines is critially involved in the recruitment and persistence of pathogenic inflammatory cells in target organs, including the skin[47], although the primary factors that control chemokine production in specific tissues are still not clear. In active AD and psoriatic lesions, many CC and CXC chemokines have been found abundantly expressed, suggesting that their presence is pivotal to disease pathogenesis[47]. Although a comprehensive picture is not clear, the chemokine signatures in human AD and psoriasis biopsies show not only molecules characteristic to each disease, but also a number that are shared between the diseases. Our data now reveal that there are factors like TWEAK that are capable of providing the stimulus for the production of many of these chemokines in the skin regardless of the underlying pathology of the specific skin disease.

TWEAK alone had some effect on chemokine production *in vitro* in keratinocytes, but this was broadened and enhanced by synergistic signals from IL-13 and IL-17, two of the primary factors critical for the pathogenesis of AD and psoriasis, respectively, suggesting the likelihood of co-operation *in vivo* between these cytokines in controlling chemokine expression. However, interestingly, a deficiency in TWEAK strongly ameliorated skin chemokine expression, suggesting its activity is dominant and not compensated by these other factors. Blocking individual chemokines therapeutically in inflammatory disorders has largely proven ineffective, likely due to redundancy of these molecules, many of which can act through the same receptor, as well as plasticity of the inflammatory milieu in complex inflammatory disorders[48,49]. But, targeting upstream signals, such as from TWEAK/Fn14, that can influence the production of multiple chemokines, could be a more attractive therapeutic option to alter the global chemokine signature associated with skin inflammatory diseases. Although we show that keratinocytes can be one source of these molecules downstream of TWEAK, there are additional possible targets of TWEAK activity that might also contribute to the chemokines induced in the skin. This will be of interest to understand in future studies. However, other reports of TWEAK in variable structural cell populations not associated with the skin, such as HUVECs, rat brain endothelial cells, murine and human renal mesangial and tubular cells, mouse synovial cells and human corneal myofibroblasts, have shown it capable of inducing CCL2 and 5, and CXCL1, 8, and 10 (refs 50–55). Collectively, this implies that TWEAK may be a strong orchestrator of chemokine production in several skin cell types, and that TWEAK may be a primary regulator of chemokine production in many tissues if it is highly upregulated during disease progression.

In addition to chemokines, we found that TWEAK activity was required *in vivo* for the production of two other mediators of atopic dermatitits or psoriasis, TSLP and IL-19, respectively. TSLP has been associated with a number of allergic Th2 responses[56], and its transgenic overexpression in keratinocytes in mice resulted in development of AD symptoms[57]. TSLP is highly expressed in the lesional skin of AD patients, as well as a related skin disease, scleroderma[58,59], and may have multiple targets on which it acts that perpetuate skin disease, including dendritic cells, mast cells, eosinophils and innate lymphoid type 2 cells[56,60]. IL-19, along with several other members of the IL-10 family, are upregulated in psoriasis lesional skin[40,61–64]. The exact role of IL-19 is not clear, but it has been suggested, in synergy with IL-20 and IL-24 that share the same receptor, to enhance the epidermal hyperplastic process, promote keratinocyte migration and drive production of several factors by keratinocytes or fibroblasts, such as anti-microbial peptides like S100A7 that are found in psoriasis as well as keratinocyte growth factor[65–67]. Our finding that TWEAK can promote the expression of both TSLP and IL-19 from keratinocytes or dermal fibroblasts provides a further mechanism by which TWEAK can contribute to the development and/or maintenance of the clinical phenotypes of both AD and psoriasis.

It is also likely that TWEAK/Fn14 interactions might promote other activities that are relevant to either AD or psoriasis. One study of human foreskin keratinocytes found that TWEAK had a mild effect in promoting their proliferation when combined with a cocktail of cytokines that included IL-17 (ref. 68), and another suggested that TWEAK signals could enhance the effect of TNF and promote a low level of apoptosis in primary human epidermal keratinocytes[69]. Both phenomena could also contribute to the disease phenotypes seen in AD and psoriasis. Possibly relevant to the latter activity, a recent report by Etemadi *et al.*[70] described a psoriasis-like disease in animals

carrying a keratinocyte-specific deficiency for *Traf2*, an adaptor molecule of a number of TNFR superfamily members, including Fn14. Interestingly, this inflammatory phenotype was driven by cell death and non-canonical NF-κB activation and was partially dependent on TNF activity. It is tempting to speculate that TWEAK may have also been active in this spontaneous disease and contributed to non-canonical NF-κB signalling in keratinocytes, and their apoptosis and activation.

In summary our data show that TWEAK activity is required for maximal clinical, histological and immunological symptoms in experimental models of AD and psoriasis. Interestingly, a recent publication also demonstrated that Fn14-deficient mice on the MRL/lpr background were protected from developing skin inflammation in this model of lupus erythematosus, and that Fn14 was highly upregulated in lesional skin from SLE patients[71]. Our data, together with these results, then suggest that TWEAK might be a ubiquitous regulator of skin inflammation regardless of the origin of the disease, and that TWEAK-blocking reagents may be useful for the treatment of other immune-driven disorders as well as AD and psoriasis that involve deregulated dermal and epidermal activity. It is further possible that the activity of TWEAK extends from the inflammatory skin diseases to other forms of skin pathology, such as those that occur from irradiation, heat damage, or trauma. In these cases, the question is whether TWEAK and/or Fn14 are upregulated in the skin. Given the notion that the stimuli for these pathologies are stress-related but generally short lived, TWEAK and Fn14 may be transiently active and in these circumstances could recruit immune cells, or promote activities in structural cells, that contribute to the resolution phase of tissue damage. It will be important in future studies to understand if the action of TWEAK and Fn14 encompasses such nonspecific skin conditions.

## Methods

**Antibodies and reagents.** All chemicals were from Sigma Aldrich (St Louis, MO), unless otherwise stated. House dust mice extract (*Dermatophagoides farinae* or Der f, XPB81D3A2.5) was from Greer Laboratories (Lenoir, NC). SEB was from Toxin Technology (Sarasota, FL). Anti-mouse TWEAK (clone mP2D10) and murine rTWEAK fusion protein were produced by Biogen[51]. Recombinant human TWEAK was from R&D Systems (Minneapolis, MN).

**Animals.** Eight- to 12-week old male mice were used. TWEAK-deficient animals were bred in house on the C57BL/6 background, and Fn14-deficient animals on a BALB/c background[51,72]. Respective wild-type littermates were used as control animals in all experiments. The NC/Nga mouse is an inbred strain from Japanese fancy mice and is an atopic-prone strain[33]. All animals were maintained in an SPF environment. All experiments were in compliance with the regulations of the La Jolla Institute for Allergy and Immunology Animal Care Committee and the Biogen Idec Institutional Animal Care and Use Committee Protocol 339–10, in accordance with guidelines of the Association for the Assessment and Accreditation of Laboratory Animal Care.

**Animal models of skin disease.** AD-like disease was induced by epicutaneous treatment with HDM extract (10 μg per mouse and treatment) and SEB (500 ng per mouse and treatment) given in 2 cycles on days 1 and 4, and 14 and 17, on the shaved and tape-stripped back skin over a 23 day period according to established protocols[26]. In some experiments, animals were treated i.p. with 200 μg anti-TWEAK or IgG (total volume 200 ul) twice weekly throughout the experiment. Psoriasis-like disease was induced by daily application of 2.5% Imiquimod cream or vehicle control on the shaved back for 8 days[35].

**Administration of rTWEAK.** Animals were treated subcutaneously in the rostral portion of the back with 75 or 200 μg of recombinant murine TWEAK as a N-terminal tagged Fc/IgG-fusion protein (thereafter called rTWEAK)[51] or control vehicle (irrelevant IgG) in 200 μl PBS, twice weekly over a 14-day period. In some experiments, animals injected with rTWEAK were treated i.p. with a combination of 50 μg hamster-α-mouse CCL2 (clone 2H5, Bioxcell, West Lebanon, NH), 50 μg rat-α-mouse CCL5 (clone 53405, R&D systems, Minneapolis, MN, USA) and 20 μg goat-α-mouse CCL7 (AF-456, R&D systems) twice weekly, or with a cocktail of the respective control antibodies twice weekly, throughout the experiment.

**Flow cytometry of cellular infiltrates from skin biopsies.** Three 6 mm punch biopsies from each animal were performed and tissues pooled for processing and staining. Skin specimens were digested using 2 mg ml$^{-1}$ Dispase II, 2 mg ml$^{-1}$ Collagenase Type IV (Worthington Biochemical, Lakewook, NJ), 200 U ml$^{-1}$ DNase I (Worthington Biochemical, Lakewook NJ), 2 mM Ca$^{2+}$Cl$_2$ and 2 mM Mg$^{2+}$Cl$_2$ for 45 min at 37 °C. Single-cell suspensions were counted in a Neubauer cell chamber and stained with the following antibodies (all from Biolegend, San Diego, CA): CD45-PerCP/Cy5.5 (clone 30-F11), CD11b-APC/Cy7 (clone M1/70), Ly6G-FITC (clone A1-8) and Siglec F-PE (clone E50-2440). Nonspecific staining was prevented with CD16/CD32 blockade (clone 24G2). Dead cells were excluded using the fixable viability dye eFluor780 or eFluor450 (eBiosciences, San Diego, CA). Flow analysis was performed on an LSR-II Flow Cytometer and analysed using Flow Jo Software. Total number of cellular infiltrates was calculated from relative frequencies.

**Histology and immunofluorescence.** Specimens from the dorsal skin were collected, formalin-fixed and paraffin-embedded. Four micrometre sections were stained with standard protocols for Hematoxylin and Eosin and Masson's Trichrome. Epidermal and dermal thickness was quantified from orthogonally embedded sections[73]. Immunofluorescence staining for Fn14 was performed on formalin-fixed and paraffin-embedded tissues using an AF647-coupled Fn14-antibody (Biolegend, Clone ITEM-4) and DAPI counterstaining. Isotype controls were run in parallel with all stainings. In addition, skin from an Fn14-deficient mouse served as a control for specific Fn14 staining. Sections were acquired on a Zeiss AxioScan Z1 slide scanner and analysed using Zen2 software.

**Quantitative PCR.** Total RNA was isolated from skin biopsies using TRIzol reagent (Invitrogen, Carlsbad, CA). cDNA was generated using an HighCapacity RT Kit (Applied Biosystems, Carlsbad, CA). Real-time PCR was performed using SYBR Green Endpoint measurement and expression calculated using the deltaCT method relative to the house-keeping gene GAPDH. Primer sequences are provided in Supplementary Table 1.

**RNAseq analysis.** WT and Fn14-deficient animals were treated with rTWEAK or IgG subcutaneously and skin biopsies were taken 3 days later. DESeq2 was used to identify differentially expressed genes. Input is the FPKM value of each gene across all samples. Values were scaled using Z-score across all samples. Fold induction of gene expression (average [AV] ± s.d.) was calculated from RNA expression in the WT rTWEAK group when compared with combined control groups (WT naive, WT IgG and Fn14$^{-/-}$ rTWEAK). Data have been deposited in NCBI's Gene Expression Omnibus and are accessible through GEO Series accession number GSE96957.

**Intracellular cytokine staining.** Draining lymph node cells were isolated from the axillary lymph nodes at the end of the experiments and stimulated *ex vivo* with 50 ng ml$^{-1}$ PMA and 1 μg ml$^{-1}$ ionomycin in the presence of Monensin for 5 h, followed by intracellular staining for IL-13-PE (eBiosciences, clone Bio13A) or IL-17A-APC/Cy7 (Biolegend, clone TC11-18H10.1). The frequency of cytokine producing CD4$^+$ T cells was determined by flow cytometry.

***In vitro* experiments using PAM212, NHEK, and NHDF cells.** The mouse keratinocyte cell line PAM212 (generous gift from Wendy Havran, Scripps Institute, La Jolla, CA) and normal human primary epidermal keratinocytes from neonates (NHEK, PS-200-010, ATCC, Manassas, VA) were grown in 10% fetal calf serum-supplemented DMEM medium and Dermal Cell Basal Medium (PCS-200-030, ATCC), respectively. Normal human dermal fibroblasts (NHDF, PCS-201-012, ATCC, Manassas, VA) were grown in 10% fetal calf serum-supplemented DMEM. Cells were cultured in 6 and 12 well format and stimulated with rTWEAK (400 ng ml$^{-1}$ murine rTWEAK for PAM212, 25-100 ng ml$^{-1}$ human rTWEAK for human cells), 100 ng ml$^{-1}$ species-specific rIL-13, 100 ng ml$^{-1}$ species-specific rIL-17A, or combinations thereof (all from Peprotech, Rocky Hill, NJ). After 24 and 48 h, cells were harvested for real-time PCR with reverse transcription or flow cytometry analysis. To measure Fn14 surface expression, cells were stained with Fn14-APC (Miltenyi, Clone ITEM-4).

**Gene set enrichment analysis.** GenePattern 2.0 (ref. 74) was used to process RNA array data for gene set enrichment analysis[75] which was run with the default settings (100 iterations, weighted). Gene enrichment was tested on human skin biopsy gene array datasets of healthy controls ($n = 8$) and patients suffering from either plaque psoriasis ($n = 9$) or intrinsic AD ($n = 7$), respectively (accession number Array Express GSE75890)[45]. A predefined set reflecting genes reported in the literature to associate with the TWEAK/Fn14 signalling pathway (Supplementary Table 2) was tested. A false discovery rate *P* value < 0.05 was considered significant.

**Statistical analysis.** Statistical analysis was performed using Prism5 software (GraphPad, San Diego, CA). Groups were analysed by *t*-test or Mann–Whitney *U*-test where indicated. A *P* value < 0.05 was considered statistically significant (*).

**Data availability.** Sequence data that support the findings of this study have been deposited in GEO and are available with the primary accession code GSE96957. The authors declare all other data supporting the findings of this study are available within the article and its Supplementary Information files.

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

## Acknowledgements

This work was supported by LJI institutional funds to M.C., and Biogen funds to L.B. D.S. was supported by a fellowship from the Swiss National Science Foundation (P300P3_155400). We thank Suzanne Szak for assistance in the RNA sequence analysis.

## Author contributions

D.S., M.C. and L.B. conceived experiments; D.S., P.W., R.H. and Me.C. performed experiments. D.W. and D.S. performed gene set enrichment analysis. D.S., L.B. and M.C. analysed the data and wrote the manuscript. Y.K., T.K. and L.B. provided animals, reagents and protocols.

## Additional information

**Competing interests:** The authors declare no competing financial interests.

**Publisher's note**: 

