## [Peer Review File · Nature Communications]

Reviewers' comments:

Reviewer #1 (Remarks to the Author):

Review 31.10.16

- You should be trying to help the work get published not necessarily in this journal but ultimately.
 - Don't criticize an experiment unless you can tell the authors how they could do it better. "If you just want to throw darts," he would say, "go to the pub."
 - Keep in mind that no one ever built a statue to a critic.
 - Try to act as a peer in the process of peer review.
- Science Signaling 2009 Michael Yaffe

Title: TWEAK Drives Inflammation in Atopic Dermatitis and Psoriasis

Manuscript #NCOMMS-16-24189

General Remarks

This study looks at the effect of depleting Fn14 on models of atopic dermatitis and psoriasis. This was a very clean very clear study, TWEAK is up regulated in these models, loss of the receptor reduces the extent of these diseases and TWAK injection recapitulates some of the phenotype. This therefore suggests that inhibiting TWEAK/Fn14 signalling could be useful in treating these diseases.

The only thing that they might want to consider in the discussion is that TWEAK induces non-canonical NF- κ B and sensitisation to TNF induced killing. The authors themselves comment on the fact that this may be keratinocyte driven. The phenotypes they see are very reminiscent of the keratinocyte deficient TRAF2 mice eLife Etemadi et al 2015 (which have constitutive non-canonical NF- κ B and are sensitive to TNF killing). If the authors have a few more words left in the discussion it might be worth making the comparison.

Reviewer #2 (Remarks to the Author):

Thank you for this interesting work looking at the role of TWEAK in the important skin diseases AD and psoriasis. The work is novel and original but I have some questions and concerns, outlined below:

(1) Your paper argues strongly that TWEAK plays a role in both AD and psoriasis, however the data may also be interpreted as showing a role in non-specific skin inflammation. It would be helpful for you to consider this possibility and if you still feel the data support specific roles in both AD and psoriasis please explain more fully how the one pathway can 'drive' what are usually mutually exclusive skin diseases in human.

(2) Your introduction and discussion raise the role of TWEAK in systemic inflammation; is there any evidence of extracutaneous involvement in the mouse models?

(3) The title is rather bold and should perhaps be re-worded to reflect the evidence presented of TWEAK maintaining/mediating inflammation (rather than 'driving') and also the fact that the data are from mouse models of AD and psoriasis.

(4) The abstract is unclear where it lists the inflammatory mediators in each disease without clearly stating which is for which disease. It is also not clear why the final statement reports

'TWEAK is a ubiquitous regulator of skin inflammation ...' this seems to be over-stating your data.

(5) The results of s/c injection of TWEAK are said to produce 'localised skin inflammation' but the distribution of inflammation is not shown in the results.

(6) What was the rationale for choosing to study the 5 chemokines (page 9, line 203)?

(7) Page 10, line 251, the data show that TWEAK plays a role in TSLP and IL-19 production but not that it is *critical*.

(8) The gene set enrichment analysis results are presented for the first time in the discussion. It would be helpful to expand this analysis (there is a considerable amount of publically available data on genetic variation and gene expression in AD and psoriasis) if possible, and this could provide support for the authors' hypothesis of a role for TWEAK in these diseases.

(9) Is there any evidence of increased or decreased apoptosis occurring within the skin of the mouse models as may be predicted from the knowledge of TWEAK?

(10) Why is there only 1 control for many of the key experiments? (Fig 1 a,b,g,h, Fig 2 a,b,g,h, Fig 4 d,f,g,h) The analyses would be more convincing if a greater number of controls could be included.

(10) In suppl fig 1, the Fn14 appears to be green (not red/brown as stated) and the lines indicating skin surface (panel B) and dermoepidermal junction (panel C) do not appear to be in the correct place.

Reply to Reviewer's critiques

Reviewer #1

We appreciate the positive comments of the reviewer.

1. The only thing that they might want to consider in the discussion is that TWEAK induces non-canonical NF- κ B and sensitisation to TNF induced killing... The phenotypes they see are very reminiscent of the keratinocyte deficient TRAF2 mice eLIFE Etemadi et al 2015 (which have constitutive non-canonical NF- κ B and are sensitive to TNF killing). If the authors have a few more words left in the discussion it might be worth making the comparison.

We have included a comment on non-canonical NF κ B signaling and apoptosis in the discussion and referenced the Etemadi paper (page 16).

Reviewer #2

Again, we are grateful for the enthusiasm from reviewer 2.

1. Your paper argues strongly that TWEAK plays a role in both AD and psoriasis, however the data may also be interpreted as showing a role in non-specific skin inflammation. It would be helpful for you to consider this possibility and if you still feel the data support specific roles in both AD and psoriasis please explain more fully how the one pathway can 'drive' what are usually mutually exclusive skin diseases in human.

To address the reviewer's comment, we have added to the discussion of the manuscript to explain more fully why we think TWEAK can be active and play an important role in what are considered two different diseases (page 13/14). We should point out there is a precedent for this in a related molecule, TNF, that is active in diseases as diverse as psoriasis, rheumatoid arthritis, and inflammatory bowel disease. The notion that AD and psoriasis are mutually exclusive diseases in humans stems from an immunological standpoint from data showing they are driven by separate lineages of T cells and other immune cell types that contribute Th2 vs. Th17 cytokines respectively. Although these drive the phenotype of each disease, this does not argue against the involvement of common factors that would co-operate with these individual T cell populations or cytokines. Our data clearly demonstrate that TWEAK is important for disease in established models of AD and psoriasis that display the Th2 (IL-13) or Th17 (IL-17) phenotypes, but in either case TWEAK and its receptor are upregulated at the same time. The rationale for the importance of these molecules is then that they are not restricted to a particular lineage of cells and do not control development of these lineages of cells, and they do not rely on these cells or their cytokines for being produced or upregulated. TWEAK alone cannot promote complete AD or psoriasis phenotypes but it can promote chemokines common to both AD and psoriasis from keratinocytes and fibroblasts, as well as characteristic cytokines of both diseases such as TSLP and IL-19. As importantly, TWEAK can synergize with the disease-specific cytokines IL-13 and IL-17 in these activities, further explaining why this one pathway can be important for pathology in two diseases thought to be fundamentally different from one another. In terms of the reviewer's reference to non-specific skin inflammation, we are not sure what is meant by this statement, but it is certainly possible that TWEAK will contribute to other skin pathologies and this is something that should be investigated in the future.

2. *Your introduction and discussion raise the role of TWEAK in systemic inflammation; is there any evidence of extracutaneous involvement in the mouse models?*

Both experimental models described induce robust and localized skin pathology on the rostral back of the animals. Nevertheless, we and others have seen a systemic immune response in both HDM and IMQ models. The HDM model induces adenopathy of the draining (axillary) lymph nodes and abundant IgE serum levels (Kawakami, *Methods Mol Biol.* 2015;1220:497-502). Similarly, the IMQ model leads to lymphadenopathy, splenomegaly and blood neutrophilia at the later stages of disease (Vinter, *Exp Dermatol.* 2016 Dec 11. doi: 10.1111/exd.13269). While we have not focused our study on systemic effects, we did observe diminished adenopathy and splenomegaly in animals deficient for TWEAK or treated with TWEAK-neutralizing reagents in the respective models. It remains to be seen if TWEAK produced in the skin can itself become systemic and then directly affect immune responses in other organs, and this would be interesting to study in the future.

3. *The title is rather bold and should perhaps be re-worded to reflect the evidence presented of TWEAK maintaining/mediating inflammation (rather than 'driving') and also the fact that the data are from mouse models of AD and psoriasis.*

As requested we have altered the title of the manuscript.

4. *The abstract is unclear where it lists the inflammatory mediators in each disease without clearly stating which is for which disease. It is also not clear why the final statement reports 'TWEAK is a ubiquitous regulator of skin inflammation ...' this seems to be over-stating your data.*

We have modified the abstract to further clarify the distinct immunological mechanisms of AD and psoriasis and state the respective cytokines involved. To prevent overstatement of the study's implication, we changed the wording in the final sentence from "ubiquitous regulator" to "critical contributor".

5. *The results of s/c injection of TWEAK are said to produce 'localised skin inflammation' but the distribution of inflammation is not shown in the results.*

We have clarified in the text that injection of rTWEAK into the rostral regions of the animals back induced a clinically defined dermatitis only in the region of injection (page 8)

6. *What was the rationale for choosing to study the 5 chemokines (page 9, line 203)?*

We chose to focus on CCL2, 5 and 7 as they are involved in the influx and maintenance of monocytes/macrophages, eosinophils, and T cells common to both diseases, and CCL17 and CCL20, which are believed to be restricted to distinct populations associated with one or the other disease, eosinophils and Th2 cells for CCL17, and Th17 cells for CCL20. This is clarified in the text on page 9.

7. *Page 10, line 251, the data show that TWEAK plays a role in TSLP and IL-19 production but not that it is *critical*.*

We have altered the text to read "played a role in" rather than "was critical for" (page 11).

8. *The gene set enrichment analysis results are presented for the first time in the discussion. It would be helpful to expand this analysis (there is a considerable amount of publically available data on genetic variation and gene expression in AD and psoriasis) if possible, and*

this could provide support for the authors' hypothesis of a role for TWEAK in these diseases.

We included the gene set enrichment analysis in the paper as we thought it was a nice addition to the publications showing upregulation of Fn14 and TWEAK at the protein level in skin or serum samples from patients with AD or psoriasis. We believe the protein data is more important than the genetic analysis, but as the reviewer thinks that both are valuable to our conclusions and hypotheses regarding the human diseases, we have moved our analysis to the end of the results section while at the same time quoting the more conventional studies of protein expression in AD and psoriasis patients (page 11/12).

9. Is there any evidence of increased or decreased apoptosis occurring within the skin of the mouse models as may be predicted from the knowledge of TWEAK?

While TWEAK alone is only a weak pro-apoptotic stimulus, it can sensitize cells to other mediators, such as TNF. As remarked by reviewer 1, a recent study by Emetadi et al described a spontaneous psoriasis-like disease in animals with a deficiency in Traf2 that involved keratinocyte cell death and was partially dependent on TNF, and in line with the reviewers request we now discuss the possible involvement of TWEAK (page 16). While our experiments did not specifically focus on apoptosis, we did not observe excessive apoptotic features within the skin of our experimental animals. Furthermore, treatment of keratinocytes or dermal fibroblasts with rTWEAK alone or with IL-13 or IL-17 did not elicit overt cellular death. However, at present we cannot make a definitive statement about TWEAK and apoptosis in the skin.

10. Why is there only 1 control for many of the key experiments? (Fig 1 a,b,g,h, Fig 2 a,b,g,h, Fig 4 d,f,g,h) The analyses would be more convincing if a greater number of controls could be included.

We have now combined independent experiments in Figure 1a,b and 2a,b and added more control mice to Figures 1f,g,h, 2f,g and 4d-h. The Figure legends have been changed accordingly where applicable.

11. In suppl fig 1, the Fn14 appears to be green (not red/brown as stated) and the lines indicating skin surface (panel B) and dermoepidermal junction (panel C) do not appear to be in the correct place.

We apologize for the errors. The figure legend has been changed accordingly and the lines for skin surface and epidermal-dermal junction redrawn.

REVIEWERS' COMMENTS:

Reviewer #2 (Remarks to the Author):

Thank you for your well-argued response to my question regarding a role for TWEAK in both AD and psoriasis.

I am sorry that you did not understand my reference to 'non-specific inflammation'. By this I mean the inflammatory response that occurs in non-atopic and non-psoriatic individuals or mouse models in response to trauma, for example tape stripping or mild excoriation. If you could demonstrate a role for TWEAK in this non-specific inflammation it would clarify whether its role is in 'any' skin inflammation or whether it is a central pathway in these two inflammatory skin diseases. If you feel this outweighs the remit of your current study that is acceptable but should be explained in the manuscript.

Thank you for modifying the title however I think it would be very helpful to state 'murine model' for clarification.

Reply to Reviewer's critiques

Reviewer 2 stated: "If you could demonstrate a role for TWEAK in this non-specific inflammation it would clarify whether its role is in 'any' skin inflammation or whether it is a central pathway in these two inflammatory skin diseases. If you feel this outweighs the remit of your current study, that is acceptable but should be explained in the manuscript"

We understand the reviewer's interest in understanding whether TWEAK is involved in wound healing or other less specific skin inflammatory diseases, but appreciate the collective decision that this should be the subject of future studies and not required for the current manuscript. We now discuss a possible broader role of TWEAK in skin inflammation at the end of the discussion section of the manuscript as requested. Reviewer 2 also wrote: "Thank you for modifying the title however I think it would be very helpful to state 'murine model' for clarification". We are happy to modify the title if the editors think this is essential. However, the title incorporates the word "experimental" and the abstract is quite specific in indicating we use murine models. Thus, we do not think it is necessary to also add murine model in the title.